# Descriptive Epidemiology of Pathogens Associated with Acute Respiratory Infection in a Community-Based Study of K–12 School Children (2015–2023)

**DOI:** 10.3390/pathogens13040340

**Published:** 2024-04-19

**Authors:** Cristalyne Bell, Maureen Goss, Derek Norton, Shari Barlow, Emily Temte, Cecilia He, Caroline Hamer, Sarah Walters, Alea Sabry, Kelly Johnson, Guanhua Chen, Amra Uzicanin, Jonathan Temte

**Affiliations:** 1Department of Family Medicine and Community Health, School of Medicine and Public Health, University of Wisconsin, Madison, WI 53706, USA; cristalyne.bell@fammed.wisc.edu (C.B.); shari.barlow@fammed.wisc.edu (S.B.); emily.temte@fammed.wisc.edu (E.T.); cecilia.he@fammed.wisc.edu (C.H.); caroline.hamer@fammed.wisc.edu (C.H.); sarah.walters@fammed.wisc.edu (S.W.); alea.sabry@fammed.wisc.edu (A.S.); kelly.johnson@fammed.wisc.edu (K.J.); jon.temte@fammed.wisc.edu (J.T.); 2Department of Biostatistics and Medical Informatics, School of Medicine and Public Health, University of Wisconsin, Madison, WI 53706, USA; dlnorton2@wisc.edu (D.N.); gchen25@wisc.edu (G.C.); 3Centers for Disease Control and Prevention, Atlanta, GA 30329, USA; aau5@cdc.gov

**Keywords:** school-based surveillance, respiratory illness, respiratory viruses, acute respiratory infection, surveillance, public health

## Abstract

School-based outbreaks often precede increased incidence of acute respiratory infections in the greater community. We conducted acute respiratory infection surveillance among children to elucidate commonly detected pathogens in school settings and their unique characteristics and epidemiological patterns. The ORegon CHild Absenteeism due to Respiratory Disease Study (ORCHARDS) is a longitudinal, laboratory-supported, school-based, acute respiratory illness (ARI) surveillance study designed to evaluate the utility of cause-specific student absenteeism monitoring for early detection of increased activity of influenza and other respiratory viruses in schools from kindergarten through 12th grade. Eligible participants with ARIs provided demographic, epidemiologic, and symptom data, along with a nasal swab or oropharyngeal specimen. Multipathogen testing using reverse-transcription polymerase chain reaction (RT-PCR) was performed on all specimens for 18 respiratory viruses and 2 atypical bacterial pathogens (*Chlamydia pneumoniae* and *Mycoplasma pneumoniae*). Between 5 January 2015 and 9 June 2023, 3498 children participated. Pathogens were detected in 2455 of 3498 (70%) specimens. Rhinovirus/enteroviruses (36%) and influenza viruses A/B (35%) were most commonly identified in positive specimens. Rhinovirus/enteroviruses and parainfluenza viruses occurred early in the academic year, followed by seasonal coronaviruses, RSV, influenza viruses A/B, and human metapneumovirus. Since its emergence in 2020, SARS-CoV-2 was detected year-round and had a higher median age than the other pathogens. A better understanding of the etiologies, presentations, and patterns of pediatric acute respiratory infections can help inform medical and public health system responses.

## 1. Introduction

Respiratory viruses impose significant burdens on communities throughout the United States [1]. Ensuing acute respiratory infections (ARIs) often result in clinical care [2] and associated expenses, missed work and school [3,4], and potential emergent care and hospitalization in severe cases. Because a variety of different respiratory pathogens may cause ARIs, laboratory-confirmed test-based surveillance remains the definitive method used to monitor and examine individual pathogens and define characteristics such as ARI-related symptoms, age distributions, and trends in seasonality [5,6,7]. Respiratory viruses commonly associated with ARIs include respiratory syncytial virus (RSV), rhino/enterovirus, and seasonal coronaviruses, although ARI surveillance efforts have historically focused on vaccine-preventable viruses such as influenza [2,6]. Establishing the presentations of common respiratory pathogens in the community—outside of inpatient care settings—is crucial to fully understand the breadth of circulating viruses and to comprehend the full spectrum of timing, symptomology, severity, and subsequent ARIs.

Schools have long been recognized as a key community congregate setting for respiratory virus transmission [8]. School-aged children play an important role in the spread of infectious pathogens within school settings, households, and the greater community. Seasonal outbreaks of influenza and other respiratory viruses in schools often predate increased ARI activity in clinics, and school-based surveillance may serve as an early warning system for impending community epidemics [2,9,10,11]. We conducted ARI surveillance among school-aged children (4–18 years) in the Oregon School District (OSD) over the course of 8½ academic years to elucidate a wider spectrum of commonly detected pathogens in school settings and their unique characteristics and epidemiological patterns.

## 2. Materials and Methods

The ORegon CHild Absenteeism due to Respiratory Disease Study (ORCHARDS) is a longitudinal, school-based, community surveillance study designed to evaluate the utility of cause-specific student absenteeism monitoring for early detection of increased activity of influenza and other respiratory viruses in schools from kindergarten through 12th grade (K–12). Because respiratory disease surveillance often depends exclusively on medically attended populations, methods that allow the assessment of community outbreaks independent of clinical data are valuable and expand the scope of investigation [5]. Since ORCHARDS has systematically used a commercial multiplex RT-PCR to detect viruses associated with ARIs in its participants, it was uniquely positioned to evaluate the prevalence of respiratory viruses in a school setting and the subsequent household transmission of viruses, primarily influenza viruses and SARS-CoV-2, among family members. The complete ORCHARDS protocol, rationale, and methodology have been described and published elsewhere [5]. ORCHARDS was funded through a cooperative agreement with the Centers for Disease Control and Prevention (CDC). The findings and conclusions in this report are those of the authors and do not necessarily represent the official position of the Centers for Disease Control and Prevention. Data collection began in January 2015, and all available data from its inception through June 2023 were included in this analysis.

### 2.1. Setting

ORCHARDS is based within the Oregon School District (OSD: Dane County) in southcentral Wisconsin. The area comprising the OSD is rural/suburban, encompasses two villages (Oregon: 11,407 residents; Brooklyn: 1521 residents), and is located in close proximity to Madison (269,196 residents). This report includes data collected continuously across 8½ years between 5 January 2015 and 9 June 2023.

### 2.2. Participants

The OSD serves 4114 K–12 students (aged 4–18 years) each year within seven schools. Any child was eligible to participate provided that they had an ARI. Children did not need to be absent from school to participate, with recruitment occurring year-round, including during school breaks and summer.

### 2.3. Definition of ARI

For the purposes of ORCHARDS eligibility, ARIs were defined as illnesses with (1) onset within 7 days; (2) at least 2 respiratory symptoms from the following list: nasal discharge, nasal obstruction, sneezing, sore throat, cough, malaise, chills, and headache; and (3) a Jackson score ≥ 2 [12]. On 2 January 2021, the eligibility criteria were expanded for participants who did not meet the second or third criteria. If the screening parent/guardian answered yes to the question ‘Do you think your child may have COVID?’ and the child was experiencing loss of smell, loss of taste, or nausea/vomiting, the household was eligible to participate. The eligibility criteria were expanded in order to capture COVID-19 infections with gastrointestinal symptom presentation, which were documented in an estimated 22.8–84.1% of pediatric COVID-19 cases [13].

### 2.4. Recruitment

The recruitment methods included a voicemail message on the absentee reporting telephone line for each school, briefly informing parents of the study opportunity and providing a contact phone number. Study staff also provided information during in-person school registration at the beginning of each school year until 2019, after which registration was moved to a virtual format. The OSD included announcements and reminders about ORCHARDS in district-wide emails to student households, and the study team prepared an annual postcard for distribution in the Fall reminding families about the study opportunity.

### 2.5. Data and Specimen Collection

From 5 January 2015 through 13 March 2020, study team members made home visits to collect demographic, epidemiologic, and symptom data (supplemental form); nasal swab (NS) specimens; and high oropharyngeal (OP) specimens. The study protocol changed on 14 March 2020 to no-contact drop off and pick up of data forms and supplies for the self-collection [14] of NS specimens and has continued in this manner.

### 2.6. Laboratory Assessment

NS specimens were tested within 2 h for influenza virus using the Quidel Sofia Influenza A + B Fluorescent Immunoassay [15] through 13 March 2020. OP specimens (until 13 March 2020) and self-collected NS specimens [14] (thereafter) were tested for influenza virus by reverse-transcription polymerase chain reaction (RT-PCR) [16] at the Wisconsin State Laboratory of Hygiene (WSLH) and for influenza virus and SARS-CoV-2 by RT-PCR (including archived specimens starting in September 2019) [17]. All OP and self-collected specimens were tested for multiple respiratory pathogens at the WSLH using the Luminex^®^ NxTAG^®^ Respiratory Pathogen Panel (RPP) [18]. Eighteen respiratory virus targets in this RPP included influenza (Flu) A matrix, FluA H1, FluA H3, FluB, RSV A, RSV B, coronavirus (CoV) 229E, CoV OC43, CoV NL63, CoV HKU1, human metapneumovirus, rhinovirus/enterovirus, adenovirus, parainfluenza virus (PIV) 1, PIV-2, PIV-3, PIV-4, and human bocavirus. The RPP did not distinguish between rhinoviruses and enteroviruses. The bocavirus target has not been validated at the Wisconsin State Laboratory of Hygiene. Thus, detections were not reported to study staff and are not included in this manuscript. Also included in the NxTAG^®^ RPP were two atypical bacterial pathogens: *Chlamydia pneumoniae* and *Mycoplasma pneumoniae*.

### 2.7. Statistical Analysis

We used a 1 July through 30 June temporal format for graphics to correspond to the seasonality of respiratory viruses. The results were described as means (±standard deviations) and medians calculated using standard statistical methods. For comparisons of median ages based on pathogens, we used the Kruskal–Wallis test. For post hoc, pairwise comparisons, we used Dunn’s test with a Benjamini–Hochberg correction of the *p*-values to maintain the false discovery rate at 5%. Analyses were performed using R, version 4.32.0.

### 2.8. Human Protections

This study was approved by the Health Sciences Institutional Review Board of the University of Wisconsin. The ORCHARDS participants provided written informed consent. We followed the STROBE reporting guidelines.

## 3. Results

### 3.1. Student Demographics and Illness Characteristics

Between 5 January 2015 and 9 June 2023, we received 3942 inquiry calls from parents or guardians and enrolled 3498 K–12 students (88.8%) who met the defined study criteria for an eligible ARI. Enrollment in ORCHARDS encompassed periods before and after the emergence of SARS-CoV-2 (Figure 1a), and 2020–2022 included school-based mitigation activities such as closure, hybrid education, physical distancing, and the required use of face masks (Figure 1b). Annual enrollment increased steadily with each year of this study but declined sharply in March 2020 following the COVID-19 pandemic-driven closure of in-person instruction.

Each year, the highest volume of calls occurred between December and March (peak season: *n* = 2357 (59.8%)), with calls dropping off considerably between May and August (*n* = 369 (9.4%)) (Figure 2). The number of calls during each peak season ranged from 68 (2020–2021) to 637 (2019–2020).

As shown in Table 1, the students ranged in age from 4 to 18 years with a median age of 10 ($\bar{x}$ = 10.2, SD = 3.52). There were slightly more males (55.9%) than females (44.1%), and the participants were primarily White and non-Hispanic (91.7%). On average, the students were enrolled 2.29 days after symptom onset. Students who reported a possible source of infection (*n* = 2158 or 61.7%) most often cited a family member (49.6%) or classmate (34.6%).

Of 3444 participants who specified their illness severity, 24.5% described their illness as mild, 65.7% as moderate, and 10.8% as severe. In the 373 self-reported severe illness episodes, influenza viruses were most commonly detected in specimens (*n* = 166; 44.5%), followed by rhino/enteroviruses (59; 15.8%), seasonal coronaviruses (24; 6.4%), and parainfluenza viruses (16; 4.3%), while 94 (25.2%) tested negative for all pathogens, similar to the rates for moderate (28.7%) and mild (32.6%) illnesses. At the time of the initial ORCHARDS visit, 75.3% of the students reported missing at least one school day due to their ARI. Most (*n* = 3077 or 88.0%) participants had not seen and did not plan to see a healthcare provider for their illness.

### 3.2. Pathogen Detection

Of 3489 nasal specimens collected (99.7% collection rate) over nine seasons, 2455 (70.3%) were positive for at least one virus, and two or more viruses were detected in 171 (4.9%) specimens. The annual test positivity (percent of specimens with at least one pathogen detected) varied over the nine seasons from 54.5% (2015–2016) to 77.4% (2019–2020). The most commonly detected viruses were rhino/enteroviruses (879, 35.8% of detections), influenza viruses (850; 34.6%), and seasonal coronaviruses (264, 10.8%) (Figure 3). Detection of atypical bacteria for *M. pneumoniae* (16, 0.7%) and *C. pneumoniae* (3, 0.1%) was uncommon.

### 3.3. Timing and Seasonality

Though most viruses circulated seasonally, the temporal peaks of virus detection varied between academic years, ranging from the week of 4 December (2022–2023) at the earliest to 10 May (2020–2021) at the latest (Figure 4). Disruptions of the timing, level, and composition of viruses occurred starting in March 2020, coinciding with the COVID-19 pandemic-related school closure (Figure 1 and Figure 4). This pattern continued through the 2020–2021 school year. Throughout this study, rhino/enteroviruses were detected year-round: on average, rhino/enteroviruses were detected during 34 of 52 calendar weeks each year. Similarly, starting on 1 September 2020, when schools re-established in-person learning, SARS-CoV-2 was detected during 22 weeks each year (Figure 5). Rhino/enteroviruses and parainfluenza viruses tended to dominate in the early weeks of the academic year, typically in September and October, as the children returned to school. Influenza viruses A and B, seasonal coronaviruses, human metapneumovirus, and RSV tended to co-occur in late autumn and winter.

The highest incidence of influenza viruses occurred during the 2019–2020 school year, just before schools shut down on 13 March 2020 to prevent the spread of SARS-CoV-2. There was minimal virus diversity during the subsequent school years until the pandemic precautions were lifted (the OSD suspended masking requirements in March of 2022), and influenza and other viruses began circulating again during the 2022–2023 school year (Figure 4).

Viruses were detected in children of all ages, but the median case-patient age varied for a number of viruses (Figure 5a,b). Seasonal coronavirus, SARS-CoV-2, influenza B virus, and rhino/enteroviruses were more common in older students (>10 years of age), and RSV, parainfluenza virus, influenza A virus, human metapneumovirus, and adenoviruses were more common in younger children (<10 years of age). Significant differences existed between the median ages of cases based on virus detection (Kruskal–Wallis chi-squared = 79.605, df = 9, *p* < 0.001). Pairwise comparisons demonstrated that the median ages of case-patients with parainfluenza virus, influenza A virus, influenza B virus, RSV, adenovirus, human metapneumovirus, rhino/enterovirus, and seasonal coronavirus infections were significantly lower than those with SARS-CoV-2; case-patients with parainfluenza virus, influenza A virus, RSV, adenovirus, or human metapneumovirus infections were significantly younger than those infected by seasonal coronavirus or rhino/enterovirus (Table 2).

### 3.4. Self-Reported Symptoms and Illness Severity

The symptom distribution varied widely between viruses, as illustrated in Table 3. The most commonly reported symptoms across all viruses were cough (82.7%), nasal congestion (80.6%), and fatigue/malaise (77.1%). The least commonly reported symptoms were conjunctivitis (4.4%), diarrhea (7.9%), and vomiting (9.6%). We excluded shortness of breath, loss of taste, and loss of smell from these rankings, as they were only documented after October 2020 in the context of the COVID-19 pandemic.

## 4. Discussion

During the study period of 8½ continuous years, which spanned the periods before and after the emergence of SARS-CoV-2, each of the 20 respiratory pathogen targets included in the commercial multiplex RT-PCR system and SARS-CoV-2/influenza RT-PCR tests used in this study (*FluA matrix*, *FluA H1*, *FluA H3*, *FluB*, *RSVA*, *RSVB*, *CoV229E*, *CoVOC43*, *CoVNL63*, *CoVHKU1*, *hMPV*, *R/E*, *Ad*, *PIV-1*, *PIV-2*, *PIV-3*, *PIV-4*, *SARS-CoV-2*, *C. pneumoniae*, and *M. pneumoniae*) were detected in participating school-aged children with ARIs. In this community-based, laboratory-supported surveillance study, rhino/enteroviruses and influenza viruses (A and B) were the most commonly detected pathogens, each with different patterns of circulation. Before the emergence of SARS-CoV-2 in our study population in March 2020, ARIs were highly seasonal, with 69.5% of illness episodes occurring between December and March from September 2015 to March 2020. This pattern changed, however, in March 2020, when schools in Dane County and across the nation were preemptively closed for in-person instruction at the start of the COVID-19 pandemic [19]; concurrently, SARS-CoV-2 was initially detected in our study population [20].

Rhino/enterovirus detections were observed at least once during each calendar week over the course of 8½ years, while influenza virus detections were observed primarily between November and March (96% of detections). Rhinovirus/enterovirus and parainfluenza viruses were observed more often at the start of the academic year, followed by influenza viruses, RSV, human metapneumovirus, and seasonal coronaviruses.

Individuals indicated their households (family members) or schools (classmates) as the likely venues of acquiring their infections in almost 85% of ARI cases for which a potential source was reported. Although we did not evaluate symptoms systematically, there was a predominance across the pathogens of reported cough, nasal congestion, and fatigue. Notably, individuals in whom SARS-CoV-2 was detected had lower rates of these symptoms. This was likely due to the expansion of the eligibility criteria to include loss of taste/smell and nausea/vomiting [13] enacted on 2 January 2021 in response to the COVID-19 pandemic. Although ARIs were associated with high rates (75%) of absenteeism, medical attendance (either realized or planned) was very low (12%). This may be related to the fact that most (90%) of the assessed ARI cases were self-reported to be of mild or moderate severity.

We noted three general patterns based on age. First, the case-patients with SARS-CoV-2 infections had the highest median age at 12.4 years (Figure 5). Second, the seasonal coronavirus and rhino/enterovirus cases were in the middle age grouping (11.2 and 10.5 years, respectively). Finally, other viruses predominated at younger ages, albeit with a great deal of overlap.

Our findings share some similarities with other recent reports. A short study that evaluated multiple pathogens during the 2022–2023 school year, irrespective of symptoms, in K–12 students and staff demonstrated a predominance of rhino/enteroviruses, followed by seasonal coronaviruses [21]. A longer-term, 5-year study of medically attended pediatric ARIs from the New Vaccine Surveillance Network showed a high prevalence of rhino/enteroviruses and moderate levels of RSV, influenza viruses, and SARS-CoV-2 in medically attended children [22]. In addition, the early arrival of rhino/enteroviruses and later occurrences of RSV and influenza viruses were demonstrated in this clinical population.

SARS-CoV-2 was initially detected in our study population in March 2020 [20] and thereafter was detected year-round. Following the pandemic-driven school closure in March 2020, participation in ORCHARDS dropped significantly. In total, 175 visits were conducted from September 2020 to August 2021, compared with an annual average of 567 in the three preceding years. This was expected as a natural product of limiting the exposure and social mixing of school-aged children and their families. Consequently, reductions in the number and diversity of detections occurred during the height of the COVID-19 pandemic (2019–2020 and 2020–2021 school years; see Figure 4) and the concomitant use of non-pharmaceutical interventions (NPIs, e.g., virtual education, block scheduling, physical distancing, and masking [23]) that were employed in the OSD for student and staff safety. We can surmise various reasons for this reduction: decreased social mixing, school- and community-based NPIs, an increase in the availability of community testing for SARS-CoV-2, and greater public awareness and sensitivity to events that might cause unnecessary risk or exposure to SARS-CoV-2. Despite a smaller sample size during this time, ORCHARDS ARI surveillance continued and served as an important indicator of persisting seasonal pathogens, such as rhino/enterovirus and seasonal coronaviruses. As NPIs were relaxed, the OSD experienced a resurgence of respiratory virus activity in autumn 2022, followed by a peak in SARS-CoV-2 cases corresponding with the Omicron surge in January 2022 [24].

### 4.1. Limitations

This study had several limitations. First, the OSD serves a suburban/rural area that is primarily White and non-Hispanic, with limited diversity based on race and ethnicity. As such, we cannot generalize these findings to more diverse populations or more densely populated settings in larger cities. Second, due to the emergence and rapid evolution of the COVID-19 pandemic, a protocol change was implemented on 13 March 2020 to allow for no-contact drop off and pick up of supplies instead of in-person home visits. As such, household member data forms were used for data collection for all participants, including the ORCHARDS children, and 37 students who participated between 12 March 2020 and 6 October 2020 were missing select data that were not collected routinely from other family members (e.g., likely source of illness). Third, our dataset consisted of school-aged children (4–18 years), and any conclusions drawn from these data are necessarily restricted to this age range. Fourth, some data were based on self-assessment (e.g., severity of illness and likely source of infection). Accordingly, mischaracterizations of severity and/or misattributions of sources of infection were possibilities that could have altered the estimates. Fifth, school attendance likely had a strong influence on student recruitment and virus detection, as families were prompted to contact study staff through the school absenteeism monitoring system. Finally, school breaks and the pandemic school closure may have played a role in the reduced number of participants during those times. Thus, milder illnesses that did not result in student absenteeism were likely underrepresented, as were illnesses occurring during school breaks.

### 4.2. Strengths/Lessons Learned

ORCHARDS offers notable strengths. First, the dataset used in this assessment was highly unique in that it captured a defined population over a long timeframe with minimal changes in the study protocol. This allowed for high consistency within the data. Second, we partnered with the WSLH for laboratory assessment and routinely used multiplex RT-PCR [18]; the latter enabled us to characterize the etiology of 55% to 77% of ARI episodes in our study participants on an annual basis. The combination of early recruitment, rapid transit of specimens, and the high functionality of the reference laboratory yielded very high overall virus detection levels (70.3%). Third, the eligibility criteria allowed a diverse set of presentations, providing a representative population of school-aged children with ARIs. This permitted a fuller assessment of community levels of respiratory virus circulation. Fourth, we attained a high level of community participation, allowing a significant sample size. Finally, the long duration of our study spanned the periods both before and after the emergence of SARS-CoV-2, with an adequate baseline study period (January 2015 to March 2020) and uninterrupted assessment across the four initial school years affected by the COVID-19 pandemic (March 2020 to June 2023).

### 4.3. Summary of Findings and Implications

A wide variety of respiratory viruses contributed to ARIs in school-aged children, most notably rhino/enterovirus and influenza. These infections appeared to be most frequently acquired from family members at home and from classmates in schools, consistent with the key role of these settings in contributing to and accelerating respiratory virus transmission within communities. There were subtle differences among viruses in terms of the ages of case-patients, seasonal timing, and symptomology. The viruses that tended to infect younger children (<10 years of age) included RSV, parainfluenza virus, influenza A virus, human metapneumovirus, and adenoviruses, while seasonal coronavirus, SARS-CoV-2, influenza B virus, and rhino/enteroviruses were more common in older students. As point-of-care testing improves and expands in clinical settings, these findings may be useful for clinicians when deciding which viruses to expect and test for. The predictability of rhino/enterovirus and SARS-CoV-2 as year-round pathogens and the smaller window of circulation for pathogens like influenza and seasonal coronaviruses (94% and 88.2% of detections occurred from November to March, respectively) can be utilized contextually by clinicians and public health professionals alike when assessing patients and community risk levels for pathogen spread and exposure. The findings of this study and the subsequent analyses highlight the ways in which a comprehensive understanding of the community-level presentations and patterns of respiratory viruses may help to inform medical and public health system responses.

## Figures and Tables

**Figure 1 pathogens-13-00340-f001:**
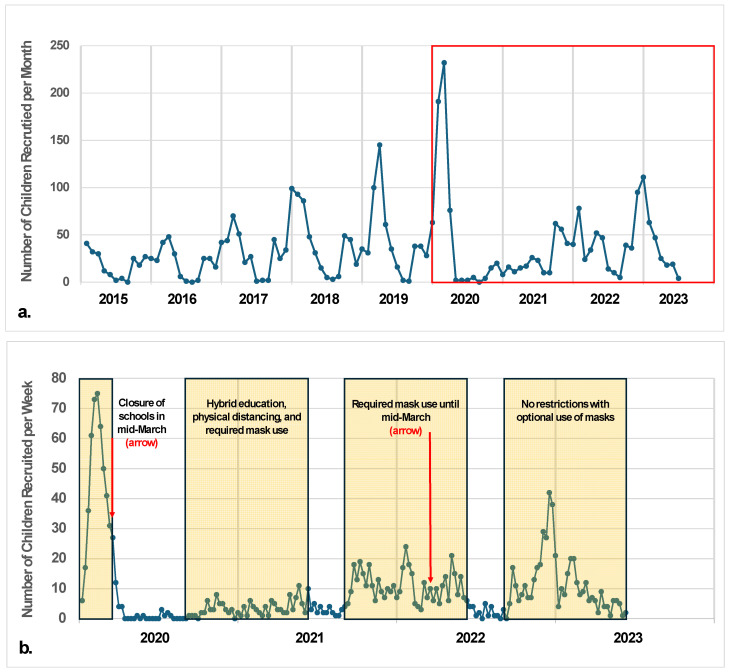
(**a**) Number of children—aged 4–18 years—recruited into the ORegon CHild Absenteeism due to Respiratory Disease Study (ORCHARDS) each month over the period from 5 January 2015 to 9 June 2023. Total participants = 3498. Box shows period from 2020 through 9 June 2023. (**b**) Detail showing number of children—aged 4–18 years—recruited into ORCHARDS each week from 2020 through 9 June 2023. Total participants in this time period = 1602. Boxes demonstrate periods when school was in session along with countermeasures employed due to the COVID-19 pandemic. Arrows depict school closure (March 2020) and end of mask requirement in schools (March 2022).

**Figure 2 pathogens-13-00340-f002:**
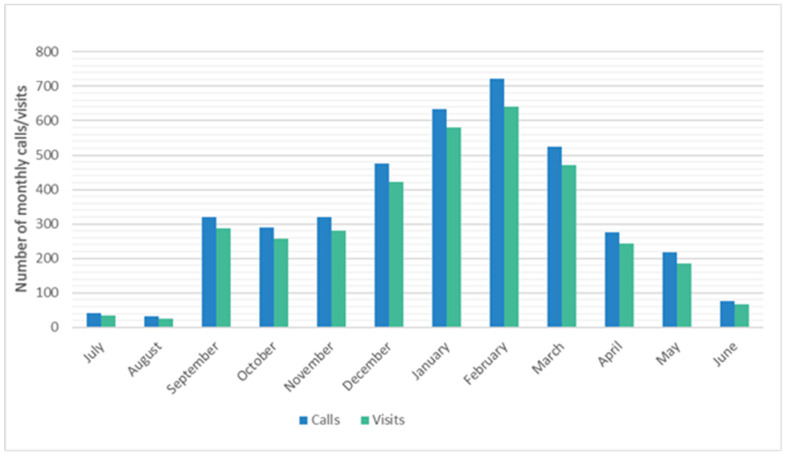
Number of enrollment inquiry calls and enrollments into the ORegon CHild Absenteeism due to Respiratory Disease Study (ORCHARDS) by month over the period from 5 January 2015 to 9 June 2023.

**Figure 3 pathogens-13-00340-f003:**
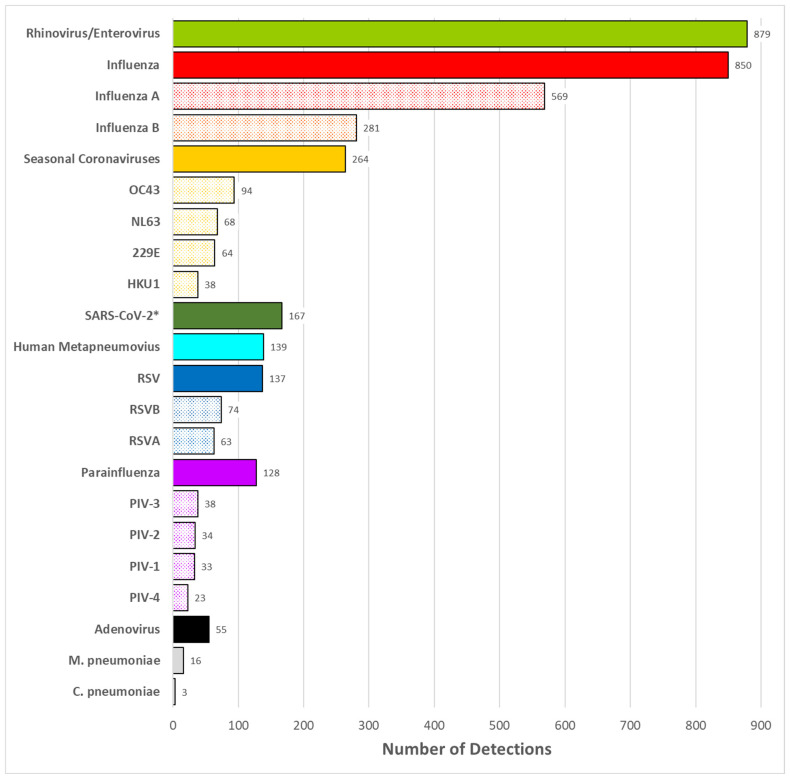
Number of pathogen detections in students from kindergarten through 12th grade in ORCHARDS over period from 5 January 2015 to 9 June 2023. Detections for virus subtypes are depicted by stippled bars. * The first detection of SARS-CoV-2 in ORCHARDS occurred on 17 March 2020; all archived specimens, starting on 1 September 2019, were tested for SARS-CoV-2.

**Figure 4 pathogens-13-00340-f004:**
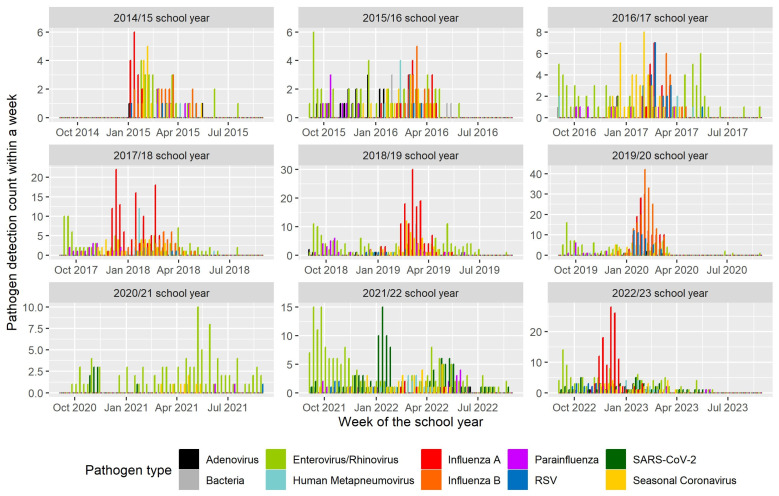
Number of respiratory pathogens detected by a commercial multiplex RT-PCR test per week across nine academic years occurring between 5 January 2015 and 9 June 2023, in students from kindergarten through 12th grade with acute respiratory infections in the Oregon School District, Dane County, Wisconsin (*n* = 3498). The bacteria included in the multiplex RT-PCR and detected in the samples were *M. pneumoniae* and *C. pneumoniae*.

**Figure 5 pathogens-13-00340-f005:**
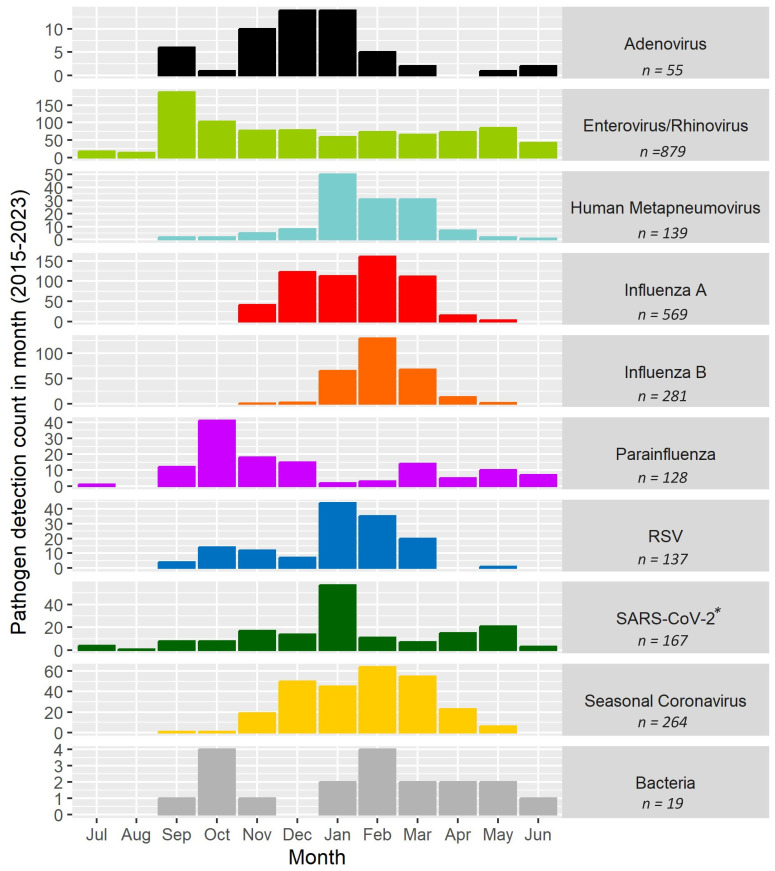
Number of respiratory pathogen detections by month from 5 January 2015 through 9 June 2023. * SARS-CoV-2 was not detected until March 2020. All strains of influenza A viruses, influenza B viruses, parainfluenza viruses, RSV, and seasonal coronaviruses are grouped together into a single category. Bacteria comprise *M. pneumoniae* and *C. pneumoniae*. (**a**) Violin plot of children’s ages for the detected respiratory pathogens. All strains of influenza A viruses, influenza B viruses, parainfluenza viruses, RSV, and seasonal coronaviruses are grouped together into a single category. Data are also shown for the bacterial pathogens *M. pneumoniae* and *C. pneumoniae*, grouped together as bacteria. (**b**) Box plot of participant age distributions for the detected respiratory pathogens, demonstrating the medians, lower quartiles, and upper quartiles of the participants’ ages associated with the detected pathogens.

**Table 1 pathogens-13-00340-t001:** Study participant demographics, rates of test positivity, reported sources of infection, and illness characteristics of the Oregon CHild Absenteeism due to Respiratory Disease Study (ORCHARDS), Dane County, Wisconsin, 2015–2023 (*n* = 3498).

	*n* (%)
Total Population	3498
Age (mean, sd *)	10.2 (3.52)
Male	1957 (55.9)
White and non-Hispanic	3175 (90.8)
Test Positivity (%)	
2014–2015	82/129 (63.6)
2015–2016	134/246 (54.5)
2016–2017	190/324 (58.6)
2017–2018	339/489 (69.3)
2018–2019	373/539 (69.2)
2019–2020	519/671 (77.4)
2020–2021	123/175 (70.3)
2021–2022	358/463 (77.3)
2022–2023	352/465 (75.7)
Reported most likely source of infection	
Family member	1070 (49.6)
School classmate	747 (34.6)
Friend	469 (21.7)
Self-reported severity of illness	
Mild	843 (24.5)
Moderate	2228 (64.7)
Severe	373 (10.8)
Length of illness before home visit (mean, sd)	2.29 (1.76)
Absent from school	2633 (75.3)
Sought or planned to seek medical care	421 (12.0)

* sd = standard deviation.

**Table 2 pathogens-13-00340-t002:** Pairwise comparison of viral pathogens and bacteria detected in study participants’ specimens. Shading indicates significant differences (*p* value ≤ 0.05).

Pathogen 1	Pathogen 2	Median Age Difference (Pathogen 1 Minus Pathogen 2)	Adjusted *p* Value
Adenovirus	Atypical Bacteria *	−2.81	0.0927
Adenovirus	Enterovirus/Rhinovirus	−2.57	0.0068
Adenovirus	Human Metapneumovirus	−1.45	0.2632
Adenovirus	Influenza A	−1.83	0.0757
Adenovirus	Influenza B	−2.26	0.0382
Adenovirus	Parainfluenza	−0.24	0.6936
Adenovirus	RSV **	−0.97	0.5135
Adenovirus	SARS-CoV-2	−4.46	0.0000
Adenovirus	Seasonal Coronavirus	−3.25	0.0013
Atypical Bacteria *	Enterovirus/Rhinovirus	0.24	0.7278
Atypical Bacteria *	Human Metapneumovirus	1.37	0.2952
Atypical Bacteria *	Influenza A	0.98	0.4085
Atypical Bacteria *	Influenza B	0.55	0.5383
Atypical Bacteria *	Parainfluenza	2.58	0.1223
Atypical Bacteria *	RSV **	1.84	0.1654
Atypical Bacteria *	SARS-CoV-2	−1.65	0.2842
Atypical Bacteria *	Seasonal Coronavirus	−0.44	0.9437
Enterovirus/Rhinovirus	Human Metapneumovirus	1.12	0.0461
Enterovirus/Rhinovirus	Influenza A	0.74	0.0260
Enterovirus/Rhinovirus	Influenza B	0.31	0.3715
Enterovirus/Rhinovirus	Parainfluenza	2.34	0.0008
Enterovirus/Rhinovirus	RSV **	1.60	0.0029
Enterovirus/Rhinovirus	SARS-CoV-2	−1.89	0.0000
Enterovirus/Rhinovirus	Seasonal Coronavirus	−0.68	0.1948
Human Metapneumovirus	Influenza A	−0.38	0.5036
Human Metapneumovirus	Influenza B	−0.81	0.2621
Human Metapneumovirus	Parainfluenza	1.21	0.3258
Human Metapneumovirus	RSV **	0.48	0.4982
Human Metapneumovirus	SARS-CoV-2	−3.01	0.0000
Human Metapneumovirus	Seasonal Coronavirus	−1.80	0.0070
Influenza A	Influenza B	−0.43	0.4511
Influenza A	Parainfluenza	1.59	0.0562
Influenza A	RSV **	0.86	0.1218
Influenza A	SARS-CoV-2	−2.63	0.0000
Influenza A	Seasonal Coronavirus	−1.42	0.0034
Influenza B	Parainfluenza	2.02	0.0223
Influenza B	RSV **	1.29	0.0507
Influenza B	SARS-CoV-2	−2.20	0.0000
Influenza B	Seasonal Coronavirus	−0.99	0.0694
Parainfluenza	RSV **	−0.73	0.7297
Parainfluenza	SARS-CoV-2	−4.22	0.0000
Parainfluenza	Seasonal Coronavirus	−3.01	0.0001
RSV **	SARS-CoV-2	−3.49	0.0000
RSV **	Seasonal Coronavirus	−2.28	0.0004
SARS-CoV-2	Seasonal Coronavirus	1.21	0.0089

* Atypical bacteria include *M. pneumoniae* and *C. pneumoniae*. ** RSV = respiratory syncytial virus.

**Table 3 pathogens-13-00340-t003:** Proportions of ARI cases with commonly reported symptoms by pathogen. Shading indicates proportions of 80% or higher.

		Self-Reported Symptoms (Proportions)						
Pathogen	Detections (*n*)	Fever	Cough	Runny Nose	Sore Throat	Fatigue	Stuffy Nose	Headache
SARS-CoV-2	167	0.49	0.34	0.33	0.34	0.41	0.35	0.30
Influenza A	570	0.83	0.93	0.72	0.68	0.86	0.79	0.66
Influenza B	283	0.87	0.94	0.75	0.75	0.92	0.79	0.71
Adenovirus	55	0.80	0.76	0.64	0.80	0.84	0.78	0.78
Coronavirus 229E	64	0.44	0.75	0.86	0.67	0.84	0.91	0.48
Coronavirus HKU1	38	0.39	0.74	0.89	0.66	0.71	0.84	0.58
Coronavirus NL63	67	0.40	0.87	0.78	0.85	0.70	0.84	0.49
Coronavirus OC43	94	0.49	0.79	0.85	0.76	0.78	0.90	0.54
Human Metapneumovirus	139	0.59	0.96	0.73	0.70	0.81	0.88	0.49
Parainfluenza 1	33	0.79	0.97	0.64	0.85	0.85	0.64	0.58
Parainfluenza 2	34	0.68	0.97	0.47	0.88	0.85	0.71	0.50
Parainfluenza 3	38	0.47	0.95	0.68	0.76	0.76	0.74	0.42
Parainfluenza 4	23	0.43	0.83	0.83	0.70	0.78	0.87	0.39
Enterovirus Rhinovirus	880	0.35	0.77	0.84	0.78	0.72	0.88	0.52
RSV * A	63	0.57	0.98	0.76	0.76	0.75	0.86	0.59
RSV * B	73	0.52	0.97	0.81	0.71	0.79	0.92	0.59
*C. pneumoniae*	3	0.67	0.67	0.67	1.00	1.00	1.00	1.00
*M. pneumoniae*	16	0.81	1.00	0.38	0.63	0.88	0.44	0.69
Totals	2641	0.57	0.83	0.75	0.72	0.77	0.81	0.56

* RSV = respiratory syncytial virus.

## Data Availability

The raw data supporting the conclusions of this article will be made available by the authors on request.

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
