# Peer review of "Descriptive Epidemiology of Pathogens Associated with Acute Respiratory Infection in a Community-Based Study of K–12 School Children (2015–2023)"

_pathogens, 2024, doi:10.3390/pathogens13040340_

Round 1

Reviewer 1 Report

Comments and Suggestions for Authors

Comments to the authors:

Summary: The paper talks about ARI surveillance among school children in the Oregon school district to elucidate commonly detected pathogens in school settings, their characteristics, and epidemiological patterns. The findings state that a wide variety of respiratory viruses contribute to ARI in school-aged children, most notably rhino/enterovirus and influenza and there were subtle differences among viruses in terms of age of case-patients, seasonal timing, and symptomology. I recommend the publication of this article after consideration of the comments below: The introduction can be improved by providing more details regarding the respiratory viruses, ARI surveillance, time period chosen for the study etc.

1.    Can authors elaborate on the respiratory viruses and acute respiratory infections in a bit more detail in the introduction.

2.    Can authors rephrase the sentence in lines 44-47 in the introduction.

3.    Can authors expand on the ARI surveillance in the introduction.

4.    Can authors provide more information on the time period chosen for this study.

5.    The rational for this study should be mentioned in the introduction clearly although a reference is mentioned (lines 72-73).

6.    Authors mentioned that the eligibility criteria were expanded in order to capture COVID-19 infections (lines 97-99). Please elaborate.

7.    Authors mentioned about pattern change before and after pandemic (lines 346-352). Can authors briefly mention about the challenges faced in the study during COVID 19 pandemic in terms of data collection and dealing with virtuality etc.

8.    Can authors elaborate on the impact of COVID on the participant number, in terms of active participation in the study.

9.    Can authors elaborate on the significance of this study in a bit more detail.

Author Response

Summary: The paper talks about ARI surveillance among school children in the Oregon school district to elucidate commonly detected pathogens in school settings, their characteristics, and epidemiological patterns. The findings state that a wide variety of respiratory viruses contribute to ARI in school-aged children, most notably rhino/enterovirus and influenza and there were subtle differences among viruses in terms of age of case-patients, seasonal timing, and symptomology. I recommend the publication of this article after consideration of the comments below: The introduction can be improved by providing more details regarding the respiratory viruses, ARI surveillance, time period chosen for the study etc.

  1. Can authors elaborate on the respiratory viruses and acute respiratory infections in a bit more detail in the introduction.
  2. Can authors rephrase the sentence in lines 44-47 in the introduction.
  3. Can authors expand on the ARI surveillance in the introduction.

To address comments 1-3 referencing the introduction, several lines have been added to afford more clarity on ARIs and specific viruses, and the sentence in lines 44-47 has been rephrased.

  1. Can authors provide more information on the time period chosen for this study.

The time period featured in this paper was not chosen, but rather is all the data available from ORCHARDS, which began collecting specimens in January of 2015 and is ongoing. This is now noted at the beginning of the methods section.

  1. The rationale for this study should be mentioned in the introduction clearly although a reference is mentioned (lines 72-73).

The rationale for this study is mentioned in the first sentence of the methods section; namely, “to evaluate the utility of cause-specific student absenteeism monitoring for early detection of increased influenza- and other respiratory virus activity in kindergarten through 12th grade (K-12) schools.” A sentence clarifying its importance has been added.

  1. Authors mentioned that the eligibility criteria were expanded in order to capture COVID-19 infections (lines 97-99). Please elaborate.

An expanded explanation has been included.

  1. Authors mentioned about pattern change before and after pandemic (lines 346-352). Can authors briefly mention about the challenges faced in the study during COVID 19 pandemic in terms of data collection and dealing with virtuality etc.
  2. Can authors elaborate on the impact of COVID on the participant number, in terms of active participation in the study.

To address comments 8 & 9, we have added language to the discussion (lines 390-410).

  1. Can authors elaborate on the significance of this study in a bit more detail.

Please see the expanded ‘Summary of findings and implications’ section for language that expounds on the study’s significance.

Reviewer 2 Report

Comments and Suggestions for Authors

Interesting work written in a good English. Methods are described well and conclusions are supported by the results.

Author Response

Thank you for taking the time to review our work! We appreciate it.

Reviewer 3 Report

Comments and Suggestions for Authors

Bell et al. conducted a community-based study spanning from 2015 to 2023 on the descriptive epidemiology of pathogens associated with acute respiratory infections in K-12 School Children. The study encompasses a significantly long timeframe, covering periods both before and after the COVID-19 pandemic. SARS-CoV-2 was also examined, including testing of archived specimens starting on September 1, 2019. The manuscript demonstrates clear and professional use of the English language. All figures are professionally drawn and easy to understand, with figure captions containing useful information for interpretation. The authors have provided a comprehensive discussion section summarizing their observations and commenting on the strengths and weaknesses of their study. Furthermore, the manuscript includes an up-to-date list of references. However, there are some suggestions to enhance the significance of this manuscript and clarify certain figures and tables:

(1) The authors should clearly state, perhaps in the discussion section, the implications for medical and public health professionals based on the observations in this study, including the types of pathogens most commonly identified, the seasonal occurrence of each disease, and the median age of patients infected by each pathogen. For instance, exploring the significance of the fact that case-patients with SARS-CoV-2 infection had the highest median age.

(2) It would be beneficial to include the sample size not only in the caption of Table 1 but also within the table itself.

(3) Figure 2 should have at least a Y-axis label for clarity.

Author Response

Bell et al. conducted a community-based study spanning from 2015 to 2023 on the descriptive epidemiology of pathogens associated with acute respiratory infections in K-12 School Children. The study encompasses a significantly long timeframe, covering periods both before and after the COVID-19 pandemic. SARS-CoV-2 was also examined, including testing of archived specimens starting on September 1, 2019. The manuscript demonstrates clear and professional use of the English language. All figures are professionally drawn and easy to understand, with figure captions containing useful information for interpretation. The authors have provided a comprehensive discussion section summarizing their observations and commenting on the strengths and weaknesses of their study. Furthermore, the manuscript includes an up-to-date list of references. However, there are some suggestions to enhance the significance of this manuscript and clarify certain figures and tables:

(1) The authors should clearly state, perhaps in the discussion section, the implications for medical and public health professionals based on the observations in this study, including the types of pathogens most commonly identified, the seasonal occurrence of each disease, and the median age of patients infected by each pathogen. For instance, exploring the significance of the fact that case-patients with SARS-CoV-2 infection had the highest median age.

Text has been added to address this comment at the end of the discussion in the section ‘Summary of findings and implications’.

(2) It would be beneficial to include the sample size not only in the caption of Table 1 but also within the table itself.

Total population has been added to the top of Table 1.

(3) Figure 2 should have at least a Y-axis label for clarity.

Y-axis label for Figure 2 has been added.